# Antispasmodic Agents in Magnetic Resonance Imaging of the Urinary Bladder—A Narrative Review

**DOI:** 10.3390/cancers16162833

**Published:** 2024-08-12

**Authors:** Katarzyna Sklinda, Martyna Rajca, Bartosz Mruk, Jerzy Walecki

**Affiliations:** 1Department of Radiology, Centre of Postgraduate Medical Education, Marymoncka 99/103, 01-813 Warsaw, Poland; 2Centre of Radiological Diagnostics, National Medical Institute of the Ministry of the Interior and Administration, Wołoska 137, 02-507 Warsaw, Poland

**Keywords:** urinary bladder MRI, VI-RADS, buscolysin, glucagon, spasmolytic effect

## Abstract

**Simple Summary:**

High-resolution magnetic resonance imaging (MRI) is essential for detecting subtle pathologies in the bladder, but it can be affected by movement from peristalsis of the urinary, gastrointestinal, and reproductive systems. Spasmolytics such as buscolysin and glucagon help to reduce motion artifacts, potentially leading to clearer images. The aim of our study was to analyze and compare the properties, benefits, and side effects of these substances to improve patient preparation for bladder MRI exams. Although both substances have similarities, butylscopolamine is preferred due to its lower cost, with glucagon serving as an alternative for those with medical contraindications. We also reviewed recent studies on the use of spasmolytics in MRI of pelvic organs. Our findings indicate that the inconsistent results regarding the utility of spasmolytics highlight the need for further research to determine whether their application truly enhances MRI accuracy and quality for the examination and staging process.

**Abstract:**

Accurate assessment of muscular layer infiltration of the urinary bladder wall is crucial for diagnostic precision and is significantly influenced, among other factors, by the elimination of motion artifacts. This review explores the potential benefits of using spasmolytic agents to achieve improved imaging results. Specifically, it examines two commonly available pharmaceutical preparations: butylscopolamine (buscolysin) and glucagon. The review highlights the similarities and differences between these agents and discusses the optimal methods of administration to enhance urinary bladder imaging. By addressing these factors, the article aims to provide insights into improving diagnostic accuracy in clinical practice.

## 1. Introduction

The objective of the magnetic resonance imaging (MRI) of the urinary bladder examination is to identify subtle pathologies, such as infiltration of the layers of the urinary bladder wall. To achieve this goal, it is necessary to perform the examination with a very high resolution, which requires, among other factors, the elimination of artifacts. One of the sources of artifacts is the peristalsis of the intestines, urinary bladder, and ureters. To minimize it during the examination, in addition to dietary adjustments and appropriate bladder filling [1], a spasmolytic agent can be administered [2,3,4]. In accordance with the VI-RADS guidelines, this is not obligatory [5]. However, similar to abdominal and other pelvic MR examinations, it is a common practice observed in radiology departments [6,7,8]. Buscolysin and glucagon, are both employed for their spasmolytic properties in MR imaging diagnostics to reduce smooth muscle motility in the gastrointestinal and urinary tract, consequently minimizing the occurrence of artifacts. To assess their efficacy, we conducted an analysis of their pharmacokinetics, emphasizing the impact of different administration routes and potential side effects.

The primary aim of this article is to provide a detailed analysis of the pharmacokinetic properties, contraindications, adverse effects, and economic considerations of glucagon and buscolysin. This is intended to offer a comprehensive context for making informed decisions when planning MRI examinations of the urinary bladder.

The secondary aim of this study is to broaden the perspective on the use of spasmolytics by conducting a comprehensive review and comparative analysis of recent studies. These studies investigate the application of spasmolytics to enhance imaging quality in MRI of the bladder as well as other pelvic organs. This approach seeks to derive conclusions that may also be applicable to bladder imaging, given the similar anatomical locations of these organs.

## 2. Buscolysin Overview

Buscolysin, also known as butylscopolamine bromide or hyoscine butylbromide, is a drug of parasympatholytic activity, similar to atropine, with faster but shorter-lasting peripheral action. It has an affinity with both nicotinic and muscarinic receptors. It is primarily used for the treatment of abdominal pain, esophageal spasms, bladder spasms, and renal colic. It can also be used to reduce motion-caused artifacts and improve magnetic resonance imaging quality for diagnostic purposes [9,10,11].

Pharmacokinetic profile: The pharmacokinetics of buscolysin vary depending on the route of administration, which can be oral (p.o.), rectal (p.r.), intramuscular (i.m.), or intravenous (i.v.), depending on the clinical indication [10,11].

The intramuscular or intravenous route allows for a faster onset of action compared to oral administration. The standard i.v. dosage for adults is 20 mg, with subsequent doses given every 30 min if necessary, and the maximum daily dose is 100 mg. For children aged 6 and above, the recommended dosage is 5–10 mg, 2–4 times a day [10,11]. The onset of action for the intravenous form is 10 min, with a time to peak of about 20–60 min and a duration of action around 2 h. After intravenous administration (100 mg as an infusion), the plasma concentration of hyoscine butylbromide rapidly declines, and the elimination half-life ranges between 1 and 5 h. It undergoes hepatic metabolism and is excreted through urine (42–61%) and feces (28–37%). The total clearance is 1.2 L/min, with 50% excreted as the unchanged drug through the kidneys. The metabolites excreted in the kidneys have minimal binding to muscarinic receptors and are not considered to contribute to the effect of hyoscine butylbromide. Importantly, being a quaternary ammonium compound, hyoscine butylbromide does not cross the blood–brain barrier and does not penetrate the central nervous system [9,10,11].

When administered orally or rectally, the bioavailability is less than 1%. Despite this, owing to its high tissue affinity for muscarinic receptors, hyoscine butylbromide remains available at the site of action in the intestine, exerting a local spasmolytic effect. Available data indicate that detectable plasma levels were achieved only after a single oral dose of 500 mg (therapeutic doses range between 30 mg–60 mg), and the maximum observed plasma concentration (Cmax) was 5 ng/mL. The peak plasma concentration is reached within 1–3 h, with low protein binding. The drug is primarily eliminated in the feces (69.7%), with minimal excretion in the urine (4.4%). Only 2.8% is eliminated via bile due to poor absorption. The elimination half-life ranges from 5 to 10 h, and the volume of distribution is 128 L. In the study by Guido N Tytgat, it is noted that hyoscine butylbromide exhibits local effects after oral intake at low doses in rats, ranging from 0.1 to about 3 mg/kg. This equates to approximately 10–210 mg/day in adults. Systemic effects of hyoscine butylbromide are anticipated when oral doses reach ≥30 mg/kg [9,12].

### 2.1. Diagnostic Usage

In practical terms, when it comes to diagnostic imaging, patients are generally administered a dosage of 20–40 mg intravenously right before the imaging process while the patient is positioned on the MR imaging table [13]. However, the study conducted on mice by Bilreiro et al. revealed that a high dose of Buscopan (hyoscine butylbromide) proved to be the most effective protocol for reducing bowel motion. This effect lasted up to 45 min, with relatively short onset timings after injection (around 8.5 min) [14]. The mice in the high-dose group received an injection of a 5 mL/kg bolus of 5 mg/kg buscolysin (1:20 dilution) intraperitoneally after 10 min of acquisition, compared to 0.5 mg/kg in the low-dose group. Significantly, there was a uniform surge in movement observed in both the buscolysin and saline groups within the initial 5 min post-injection. It was suggested that the temperature difference between the injection (approximately ${20\;}^{\circ }$C) and the body temperature of a mouse could have contributed to the short-term increase in motility. It is important to note that the study’s statistical analysis on the advantage of the high-dose group over the low-dose group did not reach statistical significance; it was considered a trend.

Further studies are required to assess the variation in human tolerance to high doses of hyoscine administered through either the intramuscular or intravenous route. Intravenous injections should be administered slowly. In children, intramuscular administration should also be done slowly, lasting at least 1 min.

### 2.2. Contraindications and Caution

Hyoscine, due to its mechanism of action, presents several contraindications. These include hypersensitivity to the drug, untreated narrow-angle glaucoma, active hemorrhage, myasthenia gravis, chronic lung disease (particularly with repeated administration), and cardiac conditions such as tachycardia, cardiac insufficiency, and angina pectoris [15,16].

Additionally, caution is advised in cases of constriction or paralysis within the gastrointestinal tract, colonic enlargement, and conditions where drug administration could lead to stagnation of gastric-intestinal contents, bloating, and poisoning.

The drug should not be administered to patients with high fever, as butylscopolamine inhibits sweat glands, potentially causing overheating. Furthermore, in elderly individuals, especially those with prostatic hyperplasia without obstruction or uropathy, caution should be exercised due to the risk of urinary retention or worsening clinical conditions [10,16,17]. This aspect is a particularly significant drawback in the context of MRI of the urinary bladder, as it often co-occurs in patients requiring bladder diagnostics. Benign prostatic hyperplasia occurs in up to 8% of the male population between the ages of 31 and 40 and even up to 90% in men in their ninth decade [18].

Interactions with butylscopolamine include enhanced anticholinergic effects when used with certain medications. Cholinomimetic and anticholinesterase drugs counteract the effects of butylscopolamine, whereas drugs such as amantadine, quinidine, tricyclic antidepressants, neuroleptics, and antihistamines enhance its muscle-relaxing properties. Butylscopolamine may interfere with the absorption and action of gastrointestinal drugs, potentially causing delays in their effects. It increases the absorption of digoxin, which raises serum concentrations and increases the risk of relative overdose. Concurrent use of corticosteroids elevates the risk of increased intraocular pressure and glaucoma. Urinary alkalizing agents may reduce the excretion of butylscopolamine, and simultaneous use with ketoconazole or metoclopramide can diminish the therapeutic effects of these drugs. Additionally, MAO inhibitors amplify both the therapeutic and side effects of butylscopolamine [10,15,17].

### 2.3. Adverse Effects

Following oral or rectal administration, common occurrences encompass hypersensitivity reactions including allergic skin symptoms such as hives, dry mouth, atonic constipation, hypotension, and an increased heart rate. Rare manifestations may include difficulty in urination and visual disturbances. Anaphylactic shock may present with an unknown frequency. Upon parenteral or intramuscular injection, various effects may arise, including increased arterial and intraocular pressure, tachycardia, heart palpitations, dizziness, headache, ataxia, dilation of the pupils, accommodation paralysis, disturbance of tear fluid secretion, thickening of respiratory secretions, dry mouth, nausea, vomiting, constipation, urinary disturbances, urinary retention, itching, rash, increased sweating, redness, fatigue, hives, and anaphylactic shock. Pain at the injection site, especially after intramuscular administration, may occur. Particularly rare occurrences, especially in children and the elderly, involve restlessness, excitement, insomnia, nervousness, and hallucinations [15,16,17,19,20,21,22,23]. We compiled the summary of buscolysin’s adverse effects and their frequency in Table 1.

### 2.4. Caution during Pregnancy

The medication may be used with special caution during pregnancy when there is a clear necessity, and the benefits of treatment, in the doctor’s opinion, outweigh the potential risk to the fetus. The drug passes into breast milk and may reduce milk secretion. Its use is considered fairly safe and compatible under certain circumstances [24].

## 3. Glucagon Overview

Glucagon, a 29-amino acid polypeptide, known for its evolutionary conservation and stability across mammalian species, activates the Gs protein, increasing cAMP concentration. Through this mechanism, it influences various metabolic processes including the metabolism of carbohydrates, proteins and lipids, hormone secretion, and inhibits peristalsis in the gastrointestinal tract [25]. It can be administered intranasally (i.n.), subcutaneously (s.c.), intramuscularly (i.m.), or intravenously (i.v.), depending on the clinical indication [26,27].

Pharmacokinetic profile: Administering glucagon subcutaneously or intramuscularly typically impacts blood glucose levels within 10 min. Intramuscular administration initiates antispasmodic action in 5–15 min, lasting for 10–40 min [27,28]. According to Gutzeit et al., intravenous administration results in a faster and more reliable onset compared to intramuscular administration [28]. The inhibitory effect on gastrointestinal motility, seen after intravenous (i.v.) administration, sets in within 1 min and lasts 5–20 min, dosage-dependent [28].

A 3 mg nasal dose raises glucose concentration after 5 min; nasal mucosal edema does not alter intranasal glucagon’s effects. Nasal administration reaches its maximum concentration (Tmax) in 15 min. The average half-life is around 38 min for the intranasal form and 3–6 min for the injection solution. Glucagon undergoes metabolism through liver, kidney, and plasma enzymes. The clearance of the glucagon is approximately 10 mL/kg/min [29]. The study by Pacchioni et al. found that both routes of administration, i.v. and i.n., were equally effective at inhibiting gastric motility [29].

### 3.1. Diagnostic Usage

Intravenous or intramuscular administration, typically 0.2–0.5 mg i.v. or 1 mg i.m., is used for relaxing the stomach, duodenal bulb, duodenum, and small intestine. For colon relaxation, usually 0.5–0.75 mg i.v. or 1–2 mg i.m. is administered. Dosage adjustments are unnecessary for elderly individuals or those with impaired kidney or liver function. The safety and efficacy of inhibiting gastrointestinal motility in children and adolescents remain undetermined. The injection solution must be stored between 2–${8\;}^{\circ }$C, with the option to store it at temperatures < ${25\;}^{\circ }$C, maintaining its shelf life for 18 months, and it should be protected from light. The nasal powder should be stored at a temperature ≤ ${30\;}^{\circ }$C [30].

### 3.2. Contraindications and Caution

Absolute contraindication to glucagon administration is limited to individuals with known hypersensitivity to the medication, with most hypersensitivity reactions arising in patients undergoing gastrointestinal imaging. Relative contraindications involve its use in neonates or children lacking sufficient glycogen stores. Patients with established insulinoma, pheochromocytoma, or a glucagon-secreting tumor are also contraindicated.

Certain glucagon formulations contain lactose, rendering them unsuitable for patients with a known lactose allergy. The injectable solution’s needle tip cover contains natural latex, posing a risk of allergic reactions in individuals allergic to latex. Glucagon’s potential to elevate myocardial oxygen demand, blood pressure, and pulse rate can be life-threatening in patients with cardiac disease, warranting cardiac monitoring during diagnostic use.

Glucagon therapy may exhibit limited efficacy in states of starvation, adrenal insufficiency, chronic hypoglycemia, and chronic alcohol abuse. To forestall hypoglycemia recurrence, patients responsive to glucagon treatment should receive oral carbohydrates to replenish hepatic glycogen deficiency.

Patients undergoing diagnostic glucagon administration may experience discomfort, particularly if fasting, manifesting as nausea, hypoglycemia, or changes in blood pressure. Post-examination, adherence to medical procedures dictates carbohydrate administration to the patient. In instances requiring fasting or severe hypoglycemia, intravenous glucose administration may become necessary. Caution is advised when employing glucagon during endoscopic and radiographic examinations in diabetic patients and elderly individuals with diagnosed heart disease [26].

Interactions with other medications include several noteworthy effects. Insulin acts antagonistically to glucagon. In individuals treated with β-adrenergic blockers, there may be enhanced circulatory reactions, such as increased heart rate and blood pressure. These reactions are transient due to glucagon’s short half-life. Glucagon may also enhance the anticoagulant effects of warfarin. When used concurrently with indomethacin, glucagon may lose its ability to increase blood glucose levels and could potentially cause hypoglycemia [27].

### 3.3. Side Effects

The injection solution may cause side effects such as nausea and vomiting, which can be observed at doses of 5 mg and above [31]. Abdominal pain is rare, and severe reactions, including hypersensitivity reactions and shock, are exceptionally rare. Adverse cardiovascular reactions are exclusive to endoscopic and radiographic examinations. For the nasal preparation, common effects include headache, increased tearing, and irritation of the upper respiratory tract. Nausea, vomiting, taste disturbances, itching and redness of the eyes, and increased blood pressure are common, while tachycardia is uncommon [27]. Overdose may lead to nausea, vomiting, weakened gastrointestinal peristalsis, increased blood pressure, and accelerated heart rate. Excessive doses may lower serum potassium levels, requiring monitoring and correction. In cases of a sudden blood pressure increase, a non-selective α-adrenergic receptor blocker is recommended. It is thought that glucagon has a less pronounced impact on cardiopulmonary function compared to butylscopolamine bromide [32]. In Table 2, we compiled the frequency of adverse effects of glucagon.

### 3.4. Caution during Pregnancy

Glucagon is considered suitable for use during pregnancy as it does not cross the placental barrier and is also safe for use during lactation. Due to its large protein structure, the concentration of glucagon in breast milk is expected to be minimal, and its absorption by the infant is unlikely because it is likely to be broken down in the gastrointestinal tract. Furthermore, glucagon has been safely administered directly to infants via injection. Consequently, no special precautions are necessary [33].

In Table 3 and Table 4, we compared selected pharmacokinetic properties of buscolysin and glucagon, as well as their contraindications.

## 4. Literature Review on Spasmolytic Use in Pelvic MRI

Among the available literature, there are unequivocal conclusions regarding the utility of administering spasmolytic agents compared in Table 5 to enhance the accuracy of MRI.

Johnson et al. investigated the impact of 20 mg of intravenous buscolysin on improving the imaging quality of pelvic organs, including the bladder, rectum, pelvic bowel, prostate, seminal vesicles, uterus, ovaries, cervix, and vagina [2]. Three radiologists evaluated MRI images before and after the administration of the spasmolytic. For all organs, the administration of buscolysin significantly improved the quality of imaging. The conclusions from this study influenced the imaging protocol standards for bladder imaging in numerous centers [3,5,8,34,35,36]. Despite the thorough analysis, the relatively small group of 47 patients and the fact that the majority of the cases were undergoing investigation for possible tumor recurrence or reassessment of tumors following treatment mean that the accuracy of the MRI assessments could not be correlated with surgical or pathological findings. Thus, it is not possible to draw a definitive conclusion that spasmolytics improve final diagnostic accuracy.

A study by A. Taylor et al. on MRI imaging in assessing the advancement of rectal cancer indicated a lack of statistically significant improvement in the effectiveness of staging evaluation compared to imaging without spasmolytics [13]. The TNM pathology results from the resected specimens served as the reference standard for evaluating the accuracy of MRI staging. The authors emphasized potential side effects and considerations of incorporating spasmolytics into diagnostics, considering the modest improvement in imaging relative to the risks associated with drug administration.

Wagner et al. led a study in which they assessed the impact of 40 mg of butylscopolamine on the quality of multiparametric MRI images (mpMRI) of the prostate [37]. They discovered that the delineation of the bowel wall significantly improved with both intramuscular and intravenous butylscopolamine administration compared to the group without spasmolytics. However, the qualitative assessment of prostate visualization, neurovascular bundle, pelvic lymph nodes, and overall image quality showed no notable differences among the groups. Therefore, butylscopolamine had a negligible effect on image quality and was considered not essential for prostate MRI.

Another research that questions the rationale for using spasmolytics is the one conducted by Sundaram et al. In their experiment, the scientists evaluated the effect of intramuscular glucagon on T2-weighted image quality in mpMRI of the prostate [38]. The results showed that although glucagon improved the quality of lymph node and rectal assessments, it did not affect the overall final evaluation of the examination or other parameters. The control group, which did not receive glucagon, achieved similar results in PI-RADS scores and biopsy yields.

The studies conducted by Ulrich et al. and Slough et al. evaluated the usefulness of hyoscine butylbromide on the quality of anatomical and functional imaging of the prostate, favoring the use of the antispasmodic agent. Both groups focused on imaging quality without examining differences and relevance for cancer detection rates [39,40].

In the study conducted by Sheikh-Sarraf et al., 20 patients were administered an intravenous injection of 1 mg/mL glucagon (GlucaGen^®^ Hypokit^®^, Novo Nordisk, Bagsværd, Denmark) to assess its potential enhancement of MR imaging [41]. The use of glucagon was correlated with a reduction in MRI artifacts (before glucagon: median 3, range 3–4; after glucagon: median 2.5, range 1–4; *p* = 0.002). This research reaffirmed the beneficial impact of the spasmolytic agent glucagon on the quality of MRI of the female pelvic region and urinary bladder [41,42,43].

Sheikh-Sarraf et al. and Cigaar et al. described a significant enhancement in image quality of the female pelvis resulting from the administration of antispasmodic agents, assessing the influence of glucagon and hyoscine, respectively [41,44]. However, neither of them compared the improvement in image quality with a change in the accuracy of staging. These studies evaluated image quality solely through a visual assessment of the presence of motion artifacts and delineation of pelvic structures.

**Table 5 cancers-16-02833-t005:** Spasmolytic use in pelvic MRI.

Authors	Assessed Organs	Substance	Sample and Methods	Results	Limitations
W. Johnson et al. [2]	Pelvic organs: bladder, rectum, pelvic bowel, prostate, seminal vesicles, uterus, ovaries cervix, vagina	20 mg i.v. buscolysin	47 patients, prospective study, paired test, 3 blinded radiologists scored overall image quality, visualization of pelvic lesions and individual pelvic organs	Scores for image quality, lesion visualization and visualization of the bladder, rectum, pelvic bowel, prostate, and seminal vesicles (all *p* < 0.0005), cervix (*p* < 0.019), and vagina (*p* < 0.0001) were significantly higher on the post-buscolysin administration imaging series (*p* < 0.0005).	The accuracy of the MRI assessments could not be correlated with the surgical or pathological findings due to the type of examined patients.
A. Taylor et al. [13]	Rectum	20 mg i.v. buscolysin	74 patients, retrospective cohort study	No statistically significant difference in overall accuracy of MRI rectal cancer staging between patients who received hyoscine butylbromide and groups who did not. No improvement in the accuracy of N-staging. The accuracy of T2 and T3 staged rectal cancers was more likely to be correct (compared with T1 cancers) with the administration of hyoscine butylbromide.	Retrospective design.
M. Wagner et al. [37]	Prostate	40 mg i.v. buscolysin, 40 mg i.m. buscolysin or none	82 patients, unpaired, retrospective, two blinded radiologists scored the visualization of the prostate capsule, central gland, and interface between the peripheral and central gland, delineation of the bowel wall, depiction of the neurovascular bundle, overall image quality	Delineation of the bowel wall on PD images was improved by both intramuscular and intravenous administration of butylscopolamine (ø–group: 3.6 0.7; i.m.–group: 2.9 0.7; i.v.–group: 2.9 0.7; *p* < 0.001). Overall image quality, quantitative evaluation of motion artefacts within the endorectal coil and ratings for depiction of different structures revealed no significant differences between the three groups.	Retrospective design: the patients underwent prostate MRI in the routine clinical setting without a strict protocol for butylscopolamine administration. The choice between intravenous and intramuscular injection was left to the radiologist performing the examination, without randomization.
K. M. Sundaram et al. [38]	Prostate	1 mg i.m. glucagon	120 patients, retrospective, three blinded radiologists assessed overall image quality, anatomic delineation (prostate capsule, rectum, and lymph nodes), and identification of benign prostatic hyperplasia nodules.	Administration of glucagon did not improve T2-weighted image quality in prostate MRI examinations and showed similar PI-RADS scores and biopsy yields compared with examinations without glucagon.	Retrospective design.
T. Ullrich et al. [39]	Prostate	40 mg i.v. buscolysin	103 patients; prospective, paired study	Hyoscine butylbromide significantly improved image quality and reduced motion-related artifacts in mpMRI of the prostate independent of bodyweight or prostate volume (*p* < 0.001). No side effects were reported.	The study was focused on the effect of buscolysin on T2-weighted imaging. Assessed image quality without investigating differences and relevance for cancer detection rates.
R. A. Slough et al. [40]	Prostate	20 mg i.v. buscolysin	173 patients, retrospective. Two blinded radiologists scored the image quality of T2-weighted imaging (T2WI), diffusion-weighted imaging (DWI), and apparent diffusion coefficient maps (ADC). DWI was further assessed for distortion and artefacts, and T2WI for the presence of motion artefacts or blurring. Dynamic contrast-enhanced image quality was assessed by recording the number of corrupt contrast curve data points	Administration of buscolysin significantly improved the image quality of T2-weighted images (*p* < 0.001). There was no significant improvement in DWI or ADC image quality, or DWI degree of distortion or artifact.	Retrospective design.
M. Sheikh-Sarraf et al. [41]	Female pelvic organs	1 mg i.v. glucagon	Prospective study performed in two centers. Two blinded radiologists scored the degradation in image quality caused by motion artifacts after the injection of glucagon.	The use of glucagon was associated with decreased MRI artifacts (before glucagon: median 3, range 3–4; after glucagon: median 2.5, range 1–4; *p* = 0.002)	Relatively small sample size. Although the *p*-value obtained in the study was significant, the strength of the effect was not large—a median of 3 vs. 2.5. The figure comparing images before and after glucagon administration was cherry-picked.
I. A. Ciggaar et al. [44]	Female pelvic organs	50 mg i.v. buscolysin	95 patients, retrospective, image quality was reviewed by visual assessment of delineation of pelvic structures (uterus, adnexa, bladder, rectum, sigmoid, uterosacral ligaments, round ligaments, small bowel) and by the presence of rectal wall edema.	Butylscopolamine provided better delineation of the small bowel and rectosigmoid compared to bisacodyl and no medication.	Retrospective design.

## 5. Discussion

Buscolysin and glucagon, despite their distinct mechanisms of action and clinical applications, exhibit commonalities in terms of pharmacokinetics and spasmolytic effects. Both substances are generally considered safe for use. However, glucagon appears to have fewer interactions and is anticipated to be a safe alternative for patients contraindicated for hyoscine. An additional critical consideration to be integrated into the decision-making process is the disparity in prices between the two substances. We compared buscolysin and glucagon products available in Poland in Table 6. In clinical practice, hyoscine is the preferred spasmolytic drug due to its significantly lower cost. Glucagon serves as an alternative for patients with contraindications to hyoscine. Particularly relevant contraindications for hyoscine in patients with bladder pathologies include prostatic hyperplasia without obstruction and uropathy.

Based on the review of studies in Section 4, it can be concluded that spasmolytics generally improve image quality, making it more visually appealing. However, the question remains whether this enhancement translates to improved accuracy in staging or characterizing lesions.

Despite the widespread use of spasmolytics in clinical MRI practice, the inconsistency of outcomes from studies conducted in different centers and on various pelvic organs indicates a need for further research to objectively assess their actual utility. Addressing this gap would enable drawing more definitive conclusions.

It is important to note the scarcity of studies specifically assessing the effect of spasmolytics on MRI of the urinary bladder. According to our research, there is only one study that thoroughly examined the impact of buscolysin on bladder MRI assessment.

## 6. Conclusions

The use of a spasmolytic agent can positively impact the diagnostic value of urinary bladder magnetic resonance imaging by reducing the influence of peristalsis, which causes motion artifacts. In most cases, butylscopolamine is the spasmolytic of choice due to its significantly lower cost and favorable pharmacokinetic parameters. It is also worth noting that in cases where patients have contraindications to the administration of butylscopolamine, a short-term beneficial spasmolytic effect can be achieved by administering glucagon.

A comprehensive understanding of the routes of administration for buscolysin and glucagon, their contraindications, associated side effects, and practical economic factors is essential for making an informed choice of antispasmodic for diagnostic purposes.

However, a review of the current literature on the use of spasmolytic agents in MRI examinations reveals inconsistent results regarding their utility in enhancing imaging accuracy and quality. It is essential to further explore the necessity of the use of spasmolytics and determine whether their application is truly beneficial for the MR imaging examination and staging process.

## 7. Future Directions

Future research should aim to standardize protocols, including the dosage and administration routes of spasmolytic agents, imaging techniques, and evaluation criteria. This would facilitate more reliable comparisons between studies and help establish clear guidelines for clinical practice.

The small number of studies specifically assessing the effect of spasmolytics on MRI of the urinary bladder indicates a significant gap in the current literature. Given the promising findings from the limited available research, further studies focusing on bladder imaging are warranted. These studies should aim to include larger patient cohorts and compare the results to the TNM pathology findings, if possible, to determine the true diagnostic benefit of spasmolytics.

## Figures and Tables

**Table 1 cancers-16-02833-t001:** Frequency of adverse effects of buscolysin [10,15].

Type of System/Organ	Adverse Effects	Incidence
Immune system disorders	Skin allergic reactions redness	1/100–1/1000, not very often
	Anaphylactic reaction	Frequency unknown ^1^
Gastro-intestinal system disorders	Mouth dryness Atonic constipation	1/100–1/1000, not very often
	Nausea Vomiting	Frequency unknown ^1^
Cardiac system disorders	Tachycardia	1/10–1/100, frequently
	Palpitations ^2^	Frequency unknown ^1^
Vascular system disorders	Hypotension	1/100–1/1000, not very often
	Hypertension ^2^	1/1000–1/10,000, rarely
Nervous system disorders	Dizziness ^2^ Paralysis of accommodation ^2^	1/10–1/100, frequently
	Disturbance of tear fluid and sweat secretion ^2^	1/100–1/1000, not very often ^1^
	Visual disturbances Headache ^2^ Ataxia ^2^ Anxiety ^2^ Agitation, insomnia ^2^ Hallucinations ^2^	1/1000–1/10,000, rarely
	Pupil dilation ^2^ Increased intraocular pressure ^2^	Frequency unknown ^1^
Urinary system disorders	Urination disorders ^2^	1/1000–1/10,000, rarely
	Urinary retention ^2^	Frequency unknown ^1^

^1^ This adverse effect was observed after the medicinal product was introduced to the market. It can be stated with 95% confidence that the frequency of occurrence is not higher than “common”, but it may be lower. Precisely estimating the frequency of occurrence is challenging due to the absence of this adverse effect in the group of 185 patients during the clinical trials. ^2^ After non-enteral administration.

**Table 2 cancers-16-02833-t002:** Frequency of adverse effects of glucagon [27].

Type of System/Organ	Adverse Effects	Incidence
Immune system disorders	Hypersensitivity reaction, including anaphylactic reaction	<1/10,000, very rarely
Metabolic system disorders	Hypoglycemia ^1^	1/100–1/1000, not very often
	Hypoglycemic coma	<1/10,000, very rarely
Cardiac system disorders	Tachycardia ^2^	<1/10,000, very rarely
Vascular system disorders	Hypotension	<1/10,000, very rarely
	Hypertension ^2^	
Gastro-intestinal system disorders	Nausea	1/10–1/100, frequently
	Vomiting	1/100–1/1000, not very often
	Abdominal pain	1/1000–1/10,000, rarely

^1^ Higher risk when administered on an empty stomach. ^2^ Increased risk in patients with pheochromocytoma, coronary artery disease, or receiving beta-blocker therapy.

**Table 3 cancers-16-02833-t003:** Comparison of selected pharmacokinetic properties of buscolysin and glucagon.

Parameter (i.v. Route)	Buscolysin	Glucagon
Dose	20–40 mg	0.2–0.75 mg
Onset of action	10 min	1 min
Time to C_max_	20–60 min	2 min
Half-life	1–5 h	3–6 min
Duration of gastrointestinal motility inhibition	45 min–2 h	5–20 min

**Table 4 cancers-16-02833-t004:** Comparison of contraindications for buscolysin and glucagon.

Buscolysin	Glucagon
Hypersensitivity Untreated narrow-angle glaucoma Myasthenia gravis Cardiac conditions (tachycardia, cardiac insufficiency, angina pectoris) Active hemorrhage Chronic lung disease Fever Hirshprung’s disease Prostatic hyperplasia without obstruction, uropathy	Hypersensitivity (additionally to lactose and latex in some formulations) Insufficient glycogen stores (neonates, starvation, adrenal insufficiency, chronic hypoglycemia, chronic alcohol abuse) Insulinoma, pheochromocytoma, glucagon-secreting tumor Cardiac disease

**Table 6 cancers-16-02833-t006:** Glucagon and buscolysin products available in Poland.

Product	Company	Dosage	Package	Full Price (PLN)	Price per mg (PLN)
**Glucagon**					
GlucaGen 1 mg HypoKit (in powder and suspension form)	Novo Nordisk, Denmark	1 mg	1 bottle + syringe	65.74	65.74
Baqsimi (nasal powder)	Eli Lilly, Netherlands	3 mg	1 dosage package	320	106.66
**Buscolysin**					
Buscolysin (solution for i.v./i. m. injection)	Sopharma, Poland	20 mg/mL	10 ampoules (1 mL each)	25.14	0.01
AuroGastro (coated tablets)	Aurovitas Pharma, Poland	10 mg	30 doses	15.30	0.1
Scopolan (coated tablets)	Herbapol, Poland	10 mg	30 doses	20.21	0.07
Scopolan (suppositories)	Herbapol, Poland	10 mg	6 doses	11.89	0.2
Buscopan (coated tablets)	Ipsen Consumer HealthCare, France	10 mg	20 doses	22.68	0.11
Buscopan Forte (coated tablets)	Ipsen Consumer HealthCare, France	20 mg	10 doses	23.91	0.12

## Data Availability

No new data were created or analyzed in this study. Data sharing is not applicable to this article.

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
