# Peer review of "Antispasmodic Agents in Magnetic Resonance Imaging of the Urinary Bladder—A Narrative Review"

_cancers, 2024, doi:10.3390/cancers16162833_

Round 1
Reviewer 1 Report
Comments and Suggestions for Authors
The paper makes a description of two drugs, its clinical use, contraindications and caution, adverse/side effects and caution during pregnancy for the two antispasmodic drugs used on MRI of urinary bladder. The description is simple but interesting considering the use.
Point 1: introduction – there are some confusion with the description of the objectives. The authors should clearly describe the primary and secondary objectives.
Point 2.1: Diagnostic Usage – in line 75 appears “Buscopan”. What this means?
In point 5. discussion, in lines 283-284, the authors refer to the chapter and transcribe its title. As there is no reference, it is assumed that they refer to the current article, under review. Neither the points are chapters, nor the title (point 4) is correct. Authors must correct.
There are four tables (5, 6, 7, 8) with the same title. The authors should consider to make a single table combining the four in only one.
References: reference 8 is badly written. The authors should review. Still about the references the authors should review all considering the journal rules.
Comments on the Quality of English LanguageNo comment.The english need to be reviewed by a native english.
Author Response
Comment 1: Point 1: introduction – there are some confusion with the description of the objectives. The authors should clearly describe the primary and secondary objectives.
Response 1: Thank you for pointing this out. We agree with this comment. Therefore, we explicitly stated the primary and secondary goals of the paper at the end of the introduction section.
Comment 2: Point 2.1: Diagnostic Usage – in line 75 appears “Buscopan”. What this means?
Response 2: Agreed. We have listed the active substance of that product in brackets right after it's mentioned.
Comment 3: In point 5. discussion, in lines 283-284, the authors refer to the chapter and transcribe its title. As there is no reference, it is assumed that they refer to the current article, under review. Neither the points are chapters, nor the title (point 4) is correct. Authors must correct.
Response 3: Agreed. We have added a hyperlink to the relevant section instead the title.
Comment 4: There are four tables (5, 6, 7, 8) with the same title. The authors should consider to make a single table combining the four in only one.
Response 4: Agreed. As the authors we are aware of this problem, but due to the lack of technical skills we are unable to render the table correctly using the MDPI latex template. We are in contact with the editorial team and trying to resolve this.
Comment 5: References: reference 8 is badly written. The authors should review. Still about the references the authors should review all considering the journal rules.
Response 5: Agreed. This was corrected, and the remaining references were double checked for errors. One overflowing url was also corrected to fit within the margins.
Thank you for the thorough review.
Reviewer 2 Report
Comments and Suggestions for Authors
The text is generally clear but could benefit from more concise phrasing in some areas. More specifically, the introduction should contain more references to strengthen the scientific credibility of the claims made. The text should flow logically from one point to the next, ensuring that related information is grouped together. The use of technical terms is appropriate, but ensure that each term is necessary and contributes to the understanding of the topic. Some sections could benefit from more explicit connections between study findings and their practical implications. For instance, while the limitations of Buscolysin and glucagon are discussed, the practical consequences for clinical practice could be more clearly stated. The work is generally well-executed, providing a thorough analysis of the use of spasmolytic agents in MRI, particularly focusing on their safety, pharmacokinetics, and potential to improve image quality. The discussion is logically structured and clearly highlights the need for further research to establish the clinical utility of these agents more definitively. Additionally, the tables provided, offer valuable information.
Comments on the Quality of English Language
The text could benefit from more concise phrasing in some areas. Some sentences are lengthy and can be split into shorter ones for better readability. There are minor grammatical errors that need correction (e.g., line 37: "Primarily used for" should be "It is primarily used for"); consistent use of punctuation, especially commas, can improve readability. Ensure the same format for dosage units and choose only one format for uniformity (e.g., "mg" vs. "milligrams" "degrees Celsius" vs."°C").
Author Response
Comment 1: the introduction should contain more references to strengthen the scientific credibility of the claims made.
Response 1: Agreed. We have added more citations to the introduction [1, 3, 4, 8]. Unfortunately we weren't able to highlight the changes due to the technicalities of latex (the references become black when highlighted).
Comment 2: The text should flow logically from one point to the next, ensuring that related information is grouped together. The use of technical terms is appropriate, but ensure that each term is necessary and contributes to the understanding of the topic.
Response 2: Thank you for pointing this out and bringing our attention to this issue. We reviewed the manuscript again and have concluded that the use of technical terms is important for the structure of the document and its meaning.
Comment 3: Some sections could benefit from more explicit connections between study findings and their practical implications. For instance, while the limitations of Buscolysin and glucagon are discussed, the practical consequences for clinical practice could be more clearly stated.
Response 3: Agreed. We have explicitly listed practical consequences in the Discussion section in the first paragraph.
Comment 4: Some sentences are lengthy and can be split into shorter ones for better readability. There are minor grammatical errors that need correction (e.g., line 37: "Primarily used for" should be "It is primarily used for"); consistent use of punctuation, especially commas, can improve readability. Ensure the same format for dosage units and choose only one format for uniformity (e.g., "mg" vs. "milligrams" "degrees Celsius" vs."°C").
Response 4: Agreed. We have reviewed and corrected the errors listed above and examined the paper for similar mistakes. We improved the consistency of the format of units.
Thank you for insightful review.